# Incidence of maternal peripartum infection: A systematic review and meta-analysis

Susannah L. Woodd[1]*, Ana Montoya[2], Maria Barreix[3], Li Pi[4], Clara Calvert[1], Andrea M. Rehman[1], Doris Chou[3], Oona M. R. Campbell[1]

1 Faculty of Epidemiology and Population Health, London School of Hygiene and Tropical Medicine, London, United Kingdom, 2 Box Hill Hospital, Eastern Health, Victoria, Australia, 3 Department of Reproductive Health and Research, World Health Organization, Geneva, Switzerland, 4 West China School of Public Health, Sichuan University, Chengdu, China

* susannah.woodd@lshtm.ac.uk

## Abstract

### Background

Infection is an important, preventable cause of maternal morbidity, and pregnancy-related sepsis accounts for 11% of maternal deaths. However, frequency of maternal infection is poorly described, and, to our knowledge, it remains the one major cause of maternal mortality without a systematic review of incidence. Our objective was to estimate the average global incidence of maternal peripartum infection.

### Methods and findings

We searched Medline, EMBASE, Global Health, and five other databases from January 2005 to June 2016 (PROSPERO: CRD42017074591). Specific outcomes comprised chorioamnionitis in labour, puerperal endometritis, wound infection following cesarean section or perineal trauma, and sepsis occurring from onset of labour until 42 days postpartum. We assessed studies irrespective of language or study design. We excluded conference abstracts, studies of high-risk women, and data collected before 1990. Three reviewers independently selected studies, extracted data, and appraised quality. Quality criteria for incidence/prevalence studies were adapted from the Joanna Briggs Institute. We used random-effects models to obtain weighted pooled estimates of incidence risk for each outcome and metaregression to identify study-level characteristics affecting incidence. From 31,528 potentially relevant articles, we included 111 studies of infection in women in labour or postpartum from 46 countries. Four studies were randomised controlled trials, two were before–after intervention studies, and the remainder were observational cohort or cross-sectional studies. The pooled incidence in high-quality studies was 3.9% (95% Confidence Interval [CI] 1.8%–6.8%) for chorioamnionitis, 1.6% (95% CI 0.9%–2.5%) for endometritis, 1.2% (95% CI 1.0%–1.5%) for wound infection, 0.05% (95% CI 0.03%–0.07%) for sepsis, and 1.1% (95% CI 0.3%–2.4%) for maternal peripartum infection. 19% of studies met all quality criteria. There were few data from developing countries and marked heterogeneity in study designs and infection definitions, limiting the interpretation of these estimates as measures

**Data Availability Statement:** Files of study characteristics and outcomes for all included studies are available from LSHTM data compass (https://doi.org/10.17037/DATA.00001411).

**Funding:** SW and AM received salary funding for this project from UNDP-UNFPA-UNICEF-WHO-World Bank Special Programme of Research, Development and Research Training in Human Reproduction (EPIDZ143). SW's salary was also supported by The Soapbox Collaborative (EPIDZB3610) http://soapboxcollaborative.org/. The funders had no role in study design, data collection and analysis, decision to publish, or preparation of the manuscript

**Competing interests:** The authors have declared that no competing interests exist.

**Abbreviations:** CI, Confidence Interval; GBD, Global Burden of Disease; KPMP, Kaiser Permanente Medical Program; LILACS, Latin American and Caribbean Health Science Information; LMICs, low- and middle-income countries; LSHTM, London School of Hygiene and Tropical Medicine; NHDS, National Hospital Discharge Survey; NHS, National Health Service; NIS, National Inpatient Sample; OR, odds ratio; SDG, sustainable development goal; SIRS, systemic inflammatory response syndrome; SSI, surgical site infection; WCC, white cell count; WHO, World Health Organization.

of global infection incidence. A limitation of this review is the inclusion of studies that were facility-based or restricted to low-risk groups of women.

## Conclusions

In this study, we observed pooled infection estimates of almost 4% in labour and between 1%–2% of each infection outcome postpartum. This indicates maternal peripartum infection is an important complication of childbirth and that preventive efforts should be increased in light of antimicrobial resistance. Incidence risk appears lower than modelled global estimates, although differences in definitions limit comparability. Better-quality research, using standard definitions, is required to improve comparability between study settings and to demonstrate the influence of risk factors and protective interventions.

## Author summary

### Why was this study done?

- Maternal infections during pregnancy and childbirth are a leading cause of preventable death in both the mother and child.

- It is unknown how frequently maternal infections occur because existing studies have not been summarised previously, to our knowledge.

- It is important for decision makers and clinical staff to know how common these infections are so that efforts are made to prevent them.

- One key reason it is difficult to summarise data on maternal infections is that the research community has used a wide variety of differing criteria to classify women as having an infection.

### What did the researchers do and find?

- We screened 31,528 research articles and included 111 in a systematic review of maternal peripartum infection, defined by the World Health Organization as infection of the genital tract and surrounding tissues during labour and up to 42 days after birth. We included articles published in all languages that would provide an estimate of the frequency of infection and found data from 46 countries.

- Using meta-analysis to combine the estimates of infection and account for variability between studies, we found that for 1,000 women giving birth, we estimated averages of 39 women with chorioamnionitis, 16 women with endometritis, 12 women with wound infection, and 0.5 women with sepsis.

- Estimates of infection varied considerably between different studies, partly explained by world region, the study design, and the criteria used to determine infection.

**What do these findings mean?**

- Infection is an important complication for many women at and after giving birth, and infection prevention should be a priority for clinicians and policymakers. However, our study found less infection than has been previously estimated.

- Representative data from all world regions were not available, highlighting knowledge gaps.

- Future research will benefit from the use of standardised infection definitions and good-quality study methods.

## Introduction

Infection is an important preventable cause of maternal morbidity and mortality, with pregnancy-related sepsis accounting for approximately 11% (95% uncertainty interval 5.9%–18.6%) of maternal deaths globally [1]. Infection also contributes significantly to deaths from other causes [2] and leads to serious consequences, including chronic pelvic inflammatory disease, ectopic pregnancy, and infertility [3]. Intrapartum fever also increases the risk of perinatal death [4]. Improved understanding of maternal infection is key to achieving the sustainable development goals (SDGs) and executing the strategies toward ending preventable maternal and neonatal mortality. However, the frequency of infection in pregnancy is poorly understood; a review of maternal morbidity identified no published systematic literature review of infection incidence, making it the one major direct cause of maternal morbidity without such a review to our knowledge [5]. A commonly cited estimate of 4% for puerperal sepsis, modelled for the 2000 Global Burden of Disease (GBD), is based on a single-centre United States (US) study, two African studies comparing home and hospital, and a Cochrane review on antibiotic prophylaxis for cesarean section comprising 66 studies [6]. Recent 2017 GBD data estimate 12.1 million incident cases of maternal sepsis and other maternal infections, including mastitis [7].

A challenge in quantifying the incidence of pregnancy-related infection is the variety of terms, definitions, time periods, sites, and severity of infections used, partly reflecting the breadth of infectious disease in this period. A commonly used term such as puerperal sepsis can range from localised symptoms and signs of genital tract infection [8] to more disseminated disease, including peritonitis, pyaemia, and sepsis [9], and with time periods that can vary from 10 days [10] to 42 days postpartum [9] and sometimes include sepsis in labour [8]. In partial response to this quantification challenge, a new definition for maternal sepsis was published in early 2018 [2]. However, the challenges remain in relation to less severe disease.

This review focusses on recent epidemiological evidence for the incidence of 'maternal peripartum infection', defined by the World Health Organization (WHO) in 2015 to encompass infections of the genital tract and surrounding tissues from onset of labour or rupture of membranes until 42 days postpartum [11]. At a time of increased global attention on maternal sepsis, this group of infections was chosen as being notable for causing over half the cases of severe maternal sepsis in the UK. In addition, the direct association of maternal peripartum infection with the process of giving birth presents key opportunities for prevention and for protecting the efficacy of antibiotics, amidst growing concerns about antimicrobial resistance

[11]. To aid prioritisation by decision makers and guide future research, we set out to estimate the average global incidence of maternal peripartum infection.

## Methods

The review was registered with PROSPERO [CRD42017074591] and conducted according to PRISMA guidelines (S1 PRISMA Checklist).

### Search strategy

We searched Medline, EMBASE, Global Health, Popline, CINAHL, the Latin American and Caribbean Health Science Information (LILACS) database, Africa-Wide Information, and regional WHO online databases using Global Index Medicus from January 2005 to June 2016. Search strategies were customised to each electronic database's individual subject headings and searching structure (S1 Text). The approach was to include articles if their abstract, title, or keywords contained a maternal term, an infection term, and a term for incidence/prevalence.

### Exclusion criteria

All identified studies were systematically assessed, irrespective of language or study design. For clinical trials in which the infection risk differed between study arms ($p < 0.05$), we used the control arm or the arm most similar to usual care. There were no case-control studies in which incidence/prevalence could be estimated.

Studies were excluded if their titles or abstracts indicated they had any of the following:

- No data on maternal peripartum infection

- A composite outcome from which it was not possible to extract data on maternal peripartum infection alone

- Only a subgroup of women at higher risk of infection than the general population of peripartum women (e.g., only cesarean section deliveries or only women with diabetes)

- No quantitative data

- No numerator

- No denominator for the total population of women

- Fewer than 30 participants

- Data collected before 1990, because of potential decreases in incidence over time. If a study spanned 1990 but disaggregated by year, data from 1990 onwards were used

- Conference and poster abstracts

- No primary data, except for reviews, which were hand-searched for additional primary studies.

We sought the full text for all remaining studies, including those for which the abstract had insufficient information to decide. The same exclusion criteria applied to full texts.

### Outcome definitions

WHO defines maternal peripartum infection as 'a bacterial infection of the genital tract or surrounding tissues occurring at any time between the onset of rupture of membranes or labour

and the 42nd day postpartum' [11]. We considered this to encompass specific constituent infections, namely chorioamnionitis in labour, puerperal endometritis, and wound infection following cesarean section, perineal tear, or episiotomy. We included sepsis occurring within the defined time period, restricted to sepsis of genital tract or wound origin when possible. We included a fifth category, 'maternal peripartum infection', for studies with a composite outcome of two or more of the above infection types or those that used a broader or unspecified definition of infection within the peripartum period.

### Measures of frequency

We aimed to estimate the incidence risk of infection in the peripartum period, defined as cases of infection emerging until 42 days postpartum among women who were infection-free at the start of labour. Because the starting point is clear (labour) and the follow-up period is short (42 days), we considered most studies to have approximated a measure of incidence risk (rather than a rate or period prevalence) and report the results as such.

### Screening and data extraction

We used the Institute of Education software, Eppi-Reviewer 4, to store citations and full-text articles, to detect duplicates, and to code screening and data extraction. SLW and AM double-screened 300 (approximately 1%) title and abstracts to ensure consistency; the rest were single-screened. Full-text screening and extraction was conducted by SLW, AM, and MB, with approximately 8% of articles double-screened and extracted to ensure consistency. AM extracted Spanish papers, and MB extracted Portuguese papers. LP screened over 40 Chinese-language papers and extracted from the included studies. Queries were resolved through discussion and, when necessary, with input from a third reviewer (OMRC). Nine authors were contacted to clarify study eligibility.

Data extracted included language, location and dates of study, study population, study design, sampling, outcome definition, denominator, time period for observing infection, data source, diagnosis, and incidence of infection (S2 Text).

### Critical appraisal of studies

We appraised the quality of each study outcome according to criteria in Table 1, adapted from Joanna Briggs Institute criteria for assessing incidence/prevalence studies [12]. For each criterion, estimates were classified as having met the criteria or not or of providing insufficient information to judge. Estimates meeting all five criteria were considered high-quality.

**Table 1. Quality assessment criteria.**

| | Quality Assessment Criteria | |
|---|---|---|
| 1 | Were study participants representative of the study target population? (appropriate recruitment strategy and sampling) | Selection bias |
| 2 | Was data analysis conducted with sufficient coverage of the identified sample? (refusals and loss are small [<15%] and unlikely to be related to the outcome) | Attrition/missing data |
| 3 | Was a clear, standard definition used for maternal infection? | Measurement bias |
| 4 | Was infection measured reliably using trained/educated data collectors, appropriate/reliable diagnostic procedures, or reliable forms of retrospective data (clinical records meeting standard definitions)? | Measurement bias |
| 5 | Were study subjects and setting described in sufficient detail to determine whether results are comparable with other studies? | Poor characterisation of study population |

To determine whether a standard definition was used (criterion 3), we compared the study definition to internationally recognised definitions for each infection (Table 2). The most recent definition of sepsis (Sepsis-3) agreed upon in early 2016 [13] and the related definition for maternal sepsis [2] proposed by WHO and JHPIEGO in 2017 were not used because these supersede our included studies; however, these revised definitions are similar to the definition for severe sepsis.

**Table 2. Standard definitions for infection outcomes.**

|  | Subgroup | Definition | Additional Comments |
|---|---|---|---|
| **Chorioamnionitis** [14] |  | Fever (>38˚C) plus one of | Studies of histological chorioamnionitis and microbial invasion of the amniotic fluid were excluded from the review. |
|  |  | a) maternal tachycardia, |  |
|  |  | b) foetal tachycardia, |  |
|  |  | c) uterine tenderness, or |  |
|  |  | d) foul-smelling vaginal discharge during labour |  |
| **Endometritis** [15] |  | At least two of the following: |  |
|  |  | a) fever (>38˚C), |  |
|  |  | b) abdominal pain with no other recognised cause, |  |
|  |  | c) uterine tenderness with no other recognised cause, or |  |
|  |  | d) purulent drainage from uterus |  |
| **Wound infection** [15] | Superficial | One of |  |
|  |  | a) purulent drainage, |  |
|  |  | b) organisms cultured, |  |
|  |  | c) incision deliberately opened AND at least one of pain, tenderness, swelling, erythema, or heat, or |  |
|  |  | d) diagnosis by attending doctor |  |
|  | Deep | Involves fascia and muscle and one of |  |
|  |  | a) purulent drainage, |  |
|  |  | b) spontaneous dehiscence or reopening AND organisms identified AND symptoms similar to superficial infection, or |  |
|  |  | c) abscess |  |
|  | Organ/space | Deeper than fascia and meets criterion for a specific organ/space infection, e.g., endometritis, and one of |  |
|  |  | a) purulent drainage from a drain, |  |
|  |  | b) organisms, or |  |
|  |  | c) abscess |  |
| **Sepsis** [16] | Infection plus SIRS | At least two of | We also accepted slightly different ranges (e.g., heart rate >100/minute, WCC >17,000/mm$^3$) because of uncertainty regarding appropriate values for pregnant and postpartum women. |
|  |  | a) temperature >38˚C or <36˚C, |  |
|  |  | b) heart rate >90/minute, |  |
|  |  | c) respiratory rate >20/minute or PaCO2 <32 mm Hg, and/or |  |
|  |  | d) WCC >12,000/mm$^3$ or <4,000/mm$^3$ or >10% immature bands |  |
|  | Severe sepsis | Sepsis associated with organ dysfunction, hypoperfusion, or hypotension. Abnormalities included, but were not limited to, lactic acidosis, oliguria, or an acute alteration in mental status | Studies that used management indicators of severe disease such as ICU admission or prolonged hospital stay were also accepted. |
|  | Blood stream infection | Positive blood culture |  |
| **Maternal peripartum infection** |  | Two or more of the above definitions, presented as a composite outcome |  |

**Abbreviations:** SIRS, systemic inflammatory response syndrome; WCC, white cell count.

If all study cases fell within these definitions, the criterion was met, even if the study definition was more restrictive and may have consequently underestimated infection incidence. Reference to national guidelines or obstetric textbooks met the criteria, as did clearly specified and appropriate ICD-9/10 codes (S1 Table). No codes exactly match the WHO definition of maternal peripartum infection, but we classified studies using ICD-9 670 (major puerperal infection, including endometritis and puerperal sepsis) [17] and ICD-10 O86 (other puerperal infection, including endometritis and wound infection) [18] as having measured maternal peripartum infection.

## Data management and analysis

We analysed infection incidence estimates separately for chorioamnionitis, endometritis, wound infection, sepsis, and maternal peripartum infection.

We exported and managed data in Microsoft Excel and STATA 15.1. We extracted information on study characteristics with potential to influence the risk of infection for use in metaregression. We categorised geographical location using SDG world regions [19]. We created a variable named 'study extent' to reflect how nationally representative the study population might be: national level (total population or representative sample), state/regional level, health facility network (e.g., surveillance network or insurance scheme), two or more facilities or field sites, and single facility or field site. Data collection was coded as routine or specific to the study. We coded diagnostic method as clinical or based on reported symptoms, except for chorioamnionitis, for which we compared the use of ICD codes with specified clinical signs. We grouped total follow-up time as being until hospital discharge, 7 days, 30 days, or 42 days postpartum. We grouped studies as only including low-risk women (e.g., low obstetric/medical risk, live birth, vaginal delivery, singleton pregnancy, or term birth) versus including all women who delivered.

We conducted meta-analyses in R version 3.5.0 using the meta [20] and metafor packages [21] to obtain a weighted pooled estimate of incidence of each infection outcome 1) for all studies, 2) for high-quality studies, and 3) stratified by world region. The pooled estimate of sepsis was also stratified by three levels of severity. When studies using nationally representative databases measured the same infection outcome over the same dates, we kept the study with the longest time period.

Infection incidence risk (as a proportion) was transformed using the Freeman–Tukey transformation to approximate a normal distribution and stabilise the variance [22, 23]. Because study designs and outcome definitions varied, we used random effects to combine study estimates [12]. The tau$^2$ measure of between-study heterogeneity was estimated using restricted maximum likelihood [24]. The pooled estimates were backtransformed, and results were presented as proportions. We generated prediction intervals to provide a predicted range for the true incidence in any individual study [25]. As sensitivity analyses, we calculated standardised residuals, removed outliers with $p > 0.05$ (based on the $t$ distribution), and noted changes in heterogeneity and prediction intervals.

We used metaregression and reported odds ratios (ORs), 95% Confidence Intervals (CIs), and $p$-values from Wald-type tests to explore whether world region or study characteristics influenced infection incidence. Infection risk was log-transformed, and univariate random-effects models were used to explore associations between each variable and odds of infection. World region and variables with evidence of association ($p < 0.1$) were included in multivariable models unless data were sparse or closely correlated.

## Results

Fig 1 shows the 31,528 potentially relevant articles identified, of which 1,543 were eligible for full-text review after title and abstract screening. We could not find two full texts. Of the

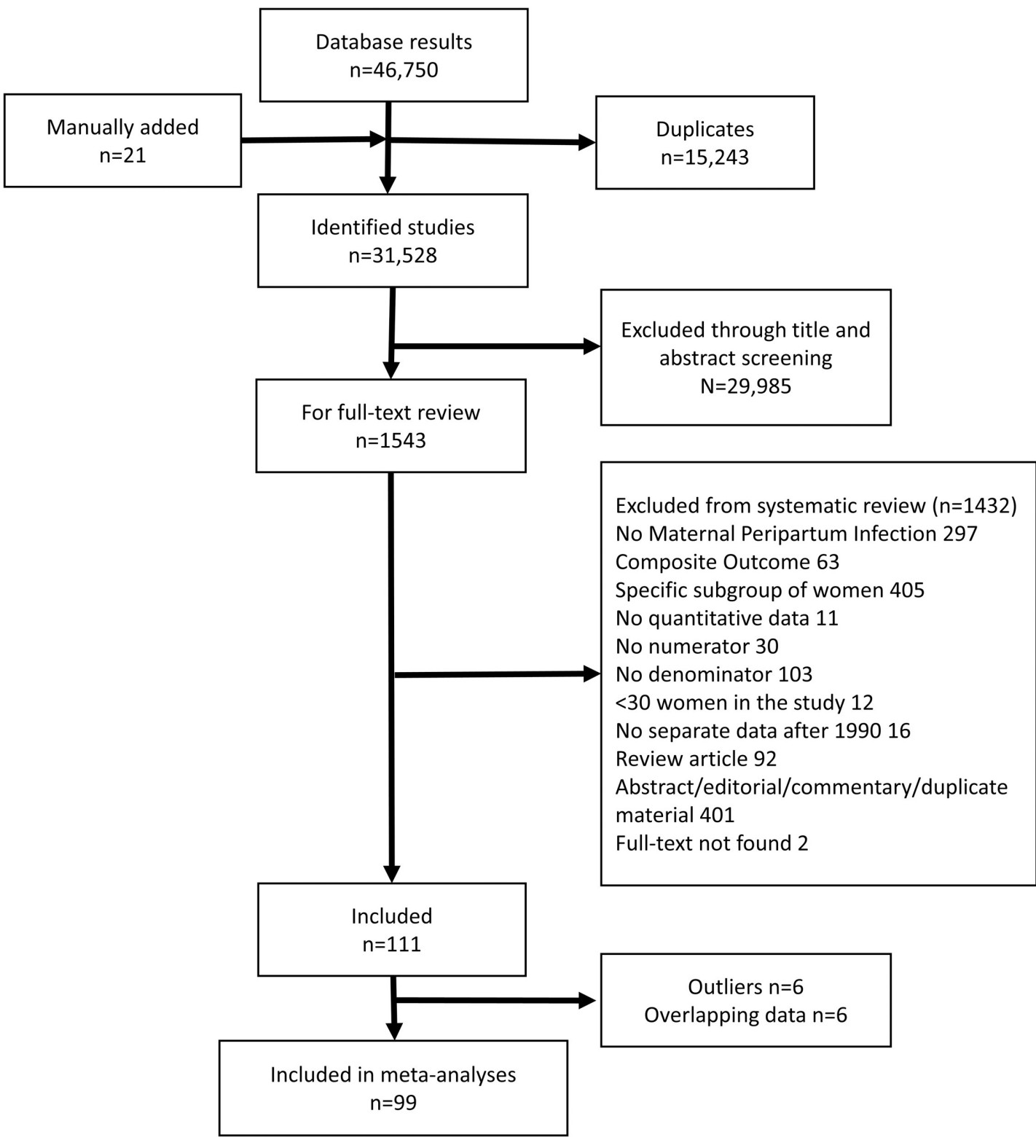

**Fig 1. Flow diagram of studies.**

remaining 1,541 full texts screened, 111 were included. Common reasons for exclusions were ineligible types of publication (N = 493) or for which the study involved only a subgroup of high-risk women (N = 405), e.g., cesarean deliveries only. Most included papers were in English, with six in Chinese [26–31], four in Spanish [32–35], four in Portuguese [36–39], three in French [40–42], and one each in Bulgarian [43], Bosnian [44], and Romanian [45]. Twenty-seven studies reported chorioamnionitis, 38 reported endometritis, 28 reported wound infection, 27 reported sepsis, and 28 reported maternal peripartum infection (S2 Table–S6 Table).

## Description of study populations

The 111 studies included data from 46 countries. Four studies were randomised controlled trials [28, 46–48], two were before–after intervention studies [27, 49], and the remainder were observational cohort or cross-sectional studies. Three studies had multiple countries: one covered nine European countries, a second involved nine Asian countries, and the third had sites in South Asia, Latin America, and sub-Saharan Africa. Of the remaining studies, 57 occurred in North America and Europe, of which 38 were in the US. There were 14 in Central and South Asia, 12 in East and Southeast Asia, 11 in Latin America, seven in sub-Saharan Africa, six in Western Asia and North Africa, and one in Australia. Nearly half the studies were conducted in one hospital, but many studies also attempted to capture all births in a country or a representative sample of them using birth certificate data or national hospital databases. In the regions/countries using such hospital databases (North America, Europe, Japan, and Thailand), over 95% of all births are in hospital facilities. In low- and middle-income countries (LMICs), only nine studies (in 10 countries: Tanzania, Nigeria, Egypt, Bangladesh, India, Pakistan, Argentina, Guatemala, Kenya, and Zambia) sought to capture population-level data.

## Study quality

Quality scores for the studies are available in S7 Table. When studies had multiple infection outcomes, the lowest score is presented. Of 111 studies, 19% met all five quality criteria, 37% met four, 22% met three, 14% met two, 7% met one, and 2% did not meet any. Only 41% of studies used a standard definition for infection, and 37% also measured infection reliably, thereby meeting both measurement criteria. In 13% of studies, there was attrition or missing data in >15% of observations, and 31% of studies had a risk of selection bias. Women or study sites were poorly characterised in 25% of studies.

## Incidence of infection

Incidence results are presented separately for the five infection outcomes (Table 3). Six studies contributed no data to the meta-analyses because of overlapping populations and dates [50–55]. Heterogeneity was high, as measured by $I^2$ (>99% for all pooled estimates), but $tau^2$ values were small and are probably more meaningful for these data since they measure actual between-study variance [56]. We identified six outlier estimates, all with high infection incidence, described below. One single-facility US study of chorioamnionitis in low-risk pregnancies provided no infection definition [57]. Three studies classified as endometritis from Bangladesh, Pakistan, and Turkey relied on self-reported symptoms of pelvic or vaginal infection [58–60]. An Indian study gave no definition for their measure of self-reported puerperal sepsis, collected up to six months after delivery [61], and similarly, a Nigerian study gave no definition for their measure of self-reported postpartum infection collected up to three years after giving birth [62]. Removal of these outliers did not change $I^2$ but led to important

**Table 3. Summary estimates for all infection outcomes.**

| | All Studies | | Meta-Analyses of All Studies (Excluding Outliers) | | | High-Quality Studies | | Meta-Analysis of High-Quality Studies | | |
|---|---|---|---|---|---|---|---|---|---|---|
| Infection Type | N | Range % | N | Pooled Incidence % (95% CI) | 95% PI | N | Range % | N | Pooled Incidence % (95% CI) | 95% PI |
| Chorioamnionitis | 28 | 0.6–19.7 | 21 | 4.1 (2.5–6.2) | 0–18.0 | 8 | 0.9–12.6 | 7 | 3.9 (1.8–6.8) | 0–17.9 |
| Endometritis | 41 | 0–16.2 | 36 | 1.4 (0.9–1.9) | 0–5.9 | 6 | 0.3–2.5 | 6 | 1.6 (0.9–2.5) | 0–6.0 |
| Wound infection | 30 | 0–10.9 | 30 | 2.1 (1.2–3.2) | 0–11.2 | 1 | 1.2 | 1 | 1.2 (1.0–1.5) | – |
| Sepsis | 31 | 0–3.8 | 26 | 0.11 (0.04–0.21) | 0–0.6 | 13 | 0.02–0.13 | 11 | 0.05 (0.03–0.07) | 0–0.18 |
| Maternal peripartum infection | 30 | 0.1–18.1 | 26 | 1.9 (1.3–2.8) | 0–7.9 | 7 | 0.2–5.8 | 7 | 1.1 (0.3–2.4) | 0–8.3 |

**Abbreviations:** CI, Confidence Interval; PI, Prediction Interval.

reductions in both tau$^2$ and prediction intervals; therefore, meta-analyses results are presented after removing these outliers.

## Chorioamnionitis

Chorioamnionitis incidence ranged from 0.6% to 19.7%, with a pooled incidence of 4.1% (95% CI 2.5%–6.2%) (Table 3). The prediction interval was wide, suggesting the incidence in any future study could lie between 0% and 18%. In North America and Europe, the pooled incidence was 4.9% (Fig 2). Only three studies were conducted in other regions. In the univariate metaregression (Table 4), study extent explained 38% of the heterogeneity, with the highest incidence seen in single-hospital studies. Studies including only singleton deliveries or only term pregnancies also had higher incidence, but almost all of these studies were conducted at single facilities.

Seven high-quality studies (meeting all five quality criteria) had a pooled infection incidence of 3.9%. The lowest incidence (0.9%) was reported in low-risk women delivering at a hospital in Bangkok, Thailand [63]. The other six estimates were from the US. Two used the US National Inpatient Sample (NIS) database and recorded a chorioamnionitis ICD-9 code in 1.7% of women in 1998–2008 [64] and 2.6% in 2008–2010 [65]. Two studies from Kaiser Permanente Medical Program (KPMP) hospitals in California also used ICD-9 codes and recorded 3.5% of women in 1995–1999 [66] and 4.0% in 2010 [67]. The highest incidences were reported in studies at single tertiary hospitals: 6.1% in Chicago [68] and 12.6% in California (among women delivering a live, single, term baby) [69].

**Endometritis.** Endometritis incidence ranged from 0%–16.2% with a pooled incidence of 1.4% (95% CI 0.9%–1.9%) (Table 3). The prediction interval suggests a true incidence of up to 6% in future studies. Pooled incidence was similar across most world regions, ranging from 1.3%–1.9%. However, it was much lower in studies from Eastern Asia and Southeastern Asia at 0.3% (Fig 3). In univariate metaregression, no variables were associated with incidence (Table 5).

Six high-quality studies had a pooled incidence of 1.6%. The lowest incidence (0.3%) was in women delivering vaginally at 66 hospitals in a surveillance network in France [70] with follow-up to 30 days postpartum. The other five studies only reported infections until hospital discharge after childbirth. Endometritis ICD-9 codes were recorded for 1.4% of women in the NIS database [65] and 1.2% of low-risk deliveries at Kaiser Permanente hospitals in California [66]. Higher infection incidence (2.4%–2.5%) was reported in three single-centre studies: two in the US [69, 71] and one in Argentina [32].

**Wound infection.** Wound infection incidence ranged from 0%–10.9%, with a pooled incidence of 2.1% (95% CI 1.2%–3.2%) (Table 3). The prediction intervals suggest the

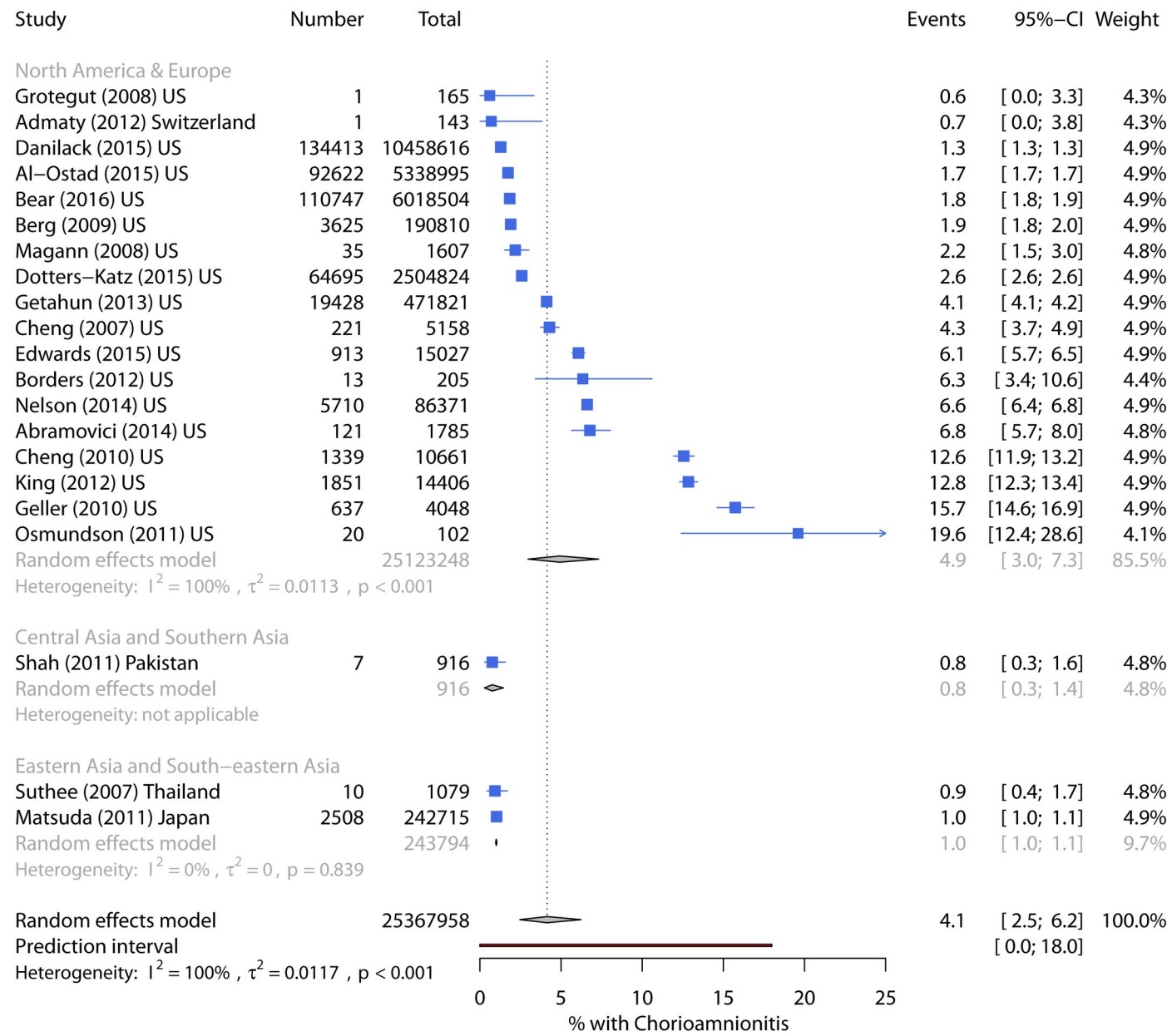

**Fig 2. Forest plot of chorioamnionitis incidence by world region.** CI, Confidence Interval.

incidence could be as high as 11.2% in future studies. Pooled incidence was highest in Eastern Asia and Southeastern Asia (6.2%) and lowest in the US and Europe (0.9%) (Fig 4). In univariate metaregression, single-site studies were associated with higher infection incidence. Unexpectedly, six studies that only included vaginal deliveries had higher pooled incidence than studies that included all delivery methods. A substantial proportion (44%) of between-study heterogeneity was explained by world region and study extent in multivariable metaregression (Table 6).

Only one study met all five quality criteria and identified 1.2% of women with cesarean or episiotomy wound infection from medical records at a single Brazilian hospital [39].

**Sepsis.** Incidence of sepsis—combining systemic inflammatory response syndrome (SIRS), severe sepsis, and blood stream infection—ranged from 0%–3.8%, with pooled

**Table 4. Chorioamnionitis univariate metaregression.**

| Factor | | No. of Studies | OR | 95% CI | *p*-Value | R² (%) |
|---|---|---|---|---|---|---|
| **Region** | North America and Europe | 18 | 1 | | | |
| | Central Asia and South Asia | 1 | 0.17 | 0.02–1.26 | | |
| | East Asia and Southeast Asia | 2 | 0.22 | 0.05–0.87 | 0.03 | 23.7 |
| **Study extent** | Single site | 12 | 1 | | | |
| | 2+ sites | 2 | 0.11 | 0.02–0.54 | | |
| | Network | 2 | 0.32 | 0.09–1.14 | | |
| | State | 1 | 0.29 | 0.05–1.58 | | |
| | National | 4 | 0.28 | 0.11–0.74 | 0.007 | 37.6 |
| **Number of foetuses** | All pregnancies | 8 | 1 | | | |
| | Singleton only | 13 | 2.64 | 1.07–6.53 | 0.04 | 13.9 |
| **Delivery mode** | All deliveries | 18 | 1 | | | |
| | Vaginal only | 3 | 1.41 | 0.37–5.43 | 0.61 | 0 |
| **Gestational age** | All gestations | 12 | 1 | | | |
| | Term only | 9 | 3.36 | 1.56–7.24 | 0.002 | 35.3 |
| **Live birth** | All deliveries | 12 | 1 | | | |
| | Live birth only | 9 | 1.16 | 0.44–3.04 | 0.77 | 0 |
| **Low risk** | All women | 16 | 1 | | | |
| | Low-risk pregnancy only | 5 | 1.56 | 0.52–4.69 | 0.43 | 0 |
| **Diagnosis** | ICD9/10 | 6 | 1 | | | |
| | Fever and other signs | 7 | 0.85 | 0.25–2.95 | | |
| | Fever only | 8 | 1.47 | 0.46–4.74 | 0.63 | 0 |
| **Data collection** | Routine | 14 | 1 | | | |
| | Study | 5 | 1.62 | 0.51–5.19 | | |
| | Unclear | 2 | 1.29 | 0.25–6.52 | 0.71 | 0 |

**Abbreviations:** CI, Confidence Interval; OR, odds ratio.

incidence 0.10% (95% CI 0.04%–0.21%) (Table 3). The prediction interval suggests the incidence could be up to 0.6% in future studies. Pooled incidence was 0.11% for SIRS, 0.08% for severe sepsis, and 0.10% for blood stream infection (S1 Fig). The majority of estimates came from the US and Europe, with a pooled incidence of 0.10%. Latin America had a similar incidence of 0.08%, whilst Central and South Asia had slightly more infection (0.27%) (Fig 5). In univariate analysis, there was weak evidence for an association with world region, no evidence for an association with severity, but increased incidence of sepsis with longer follow-up. Women with singleton pregnancies had higher infection incidence, but the two studies involved also had longer follow-up periods. Data were too sparse to investigate other factors or conduct multivariable metaregression (Table 7).

Eleven high-quality estimates produced a pooled incidence of 0.05%. Four high-quality estimates of SIRS used data from the delivery admission: NIS (0.03%) [72], all Californian hospitals (0.10%) [73], all hospitals in Thailand (0.13%) [74], and one reference hospital in São Paolo, Brazil (0.04%) [37]. Incidence of severe sepsis with organ dysfunction was low: NIS (0.01%) [72], Californian hospitals (0.05%) [73], and no cases in a near-miss study at one hospital in Gabon [41]. US data from NIS and the National Hospital Discharge Survey (NHDS) estimated blood stream infection at 0.02% [65] and 0.07% [75]. One region in Denmark and two hospitals in Ireland followed women until 30 and 42 days postpartum and identified blood stream infection in 0.06% [76] and 0.11% [77], respectively.

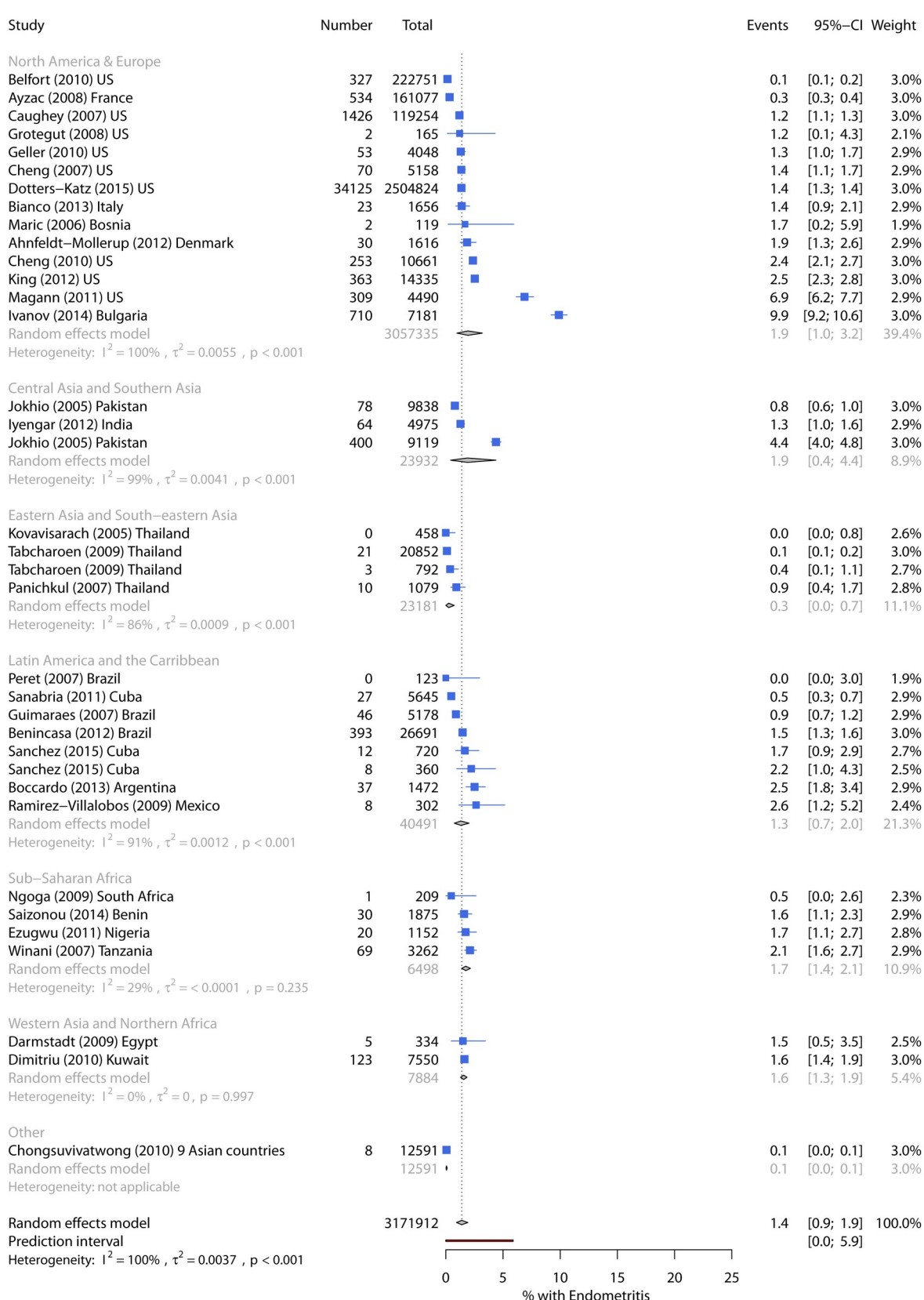

**Fig 3. Forest plot of endometritis incidence by world region.** CI, Confidence Interval.

**Table 5. Endometritis metaregression.**

| Factor | | No. of Studies | OR | 95% CI | *p*-Value | R² (%) |
|---|---|---|---|---|---|---|
| **Region** | North America and Europe | 14 | 1 | | | |
| | Central Asia and South Asia | 3 | 1.09 | 0.35–3.46 | | |
| | East Asia and Southeast Asia | 4 | 0.18 | 0.06–0.59 | | |
| | Latin America & Caribbean | 8 | 0.91 | 0.39–2.11 | | |
| | Sub-Saharan Africa | 4 | 0.99 | 0.33–2.97 | | |
| | West Asia and North Africa | 2 | 1.03 | 0.25–4.29 | 0.12 | 8.0 |
| **Study extent** | Single site | 25 | 1 | | | |
| | 2+ sites | 4 | 1.82 | 0.66–4.99 | | |
| | Network | 2 | 0.48 | 0.13–1.81 | | |
| | State | 2 | 1.44 | 0.38–5.51 | | |
| | National | 2 | 0.34 | 0.09–1.29 | 0.20 | 6.9 |
| **Number of foetuses** | All pregnancies | 23 | 1 | | | |
| | Singleton only | 12 | 1.52 | 0.75–3.07 | 0.24 | 2.6 |
| **Delivery mode** | All deliveries | 31 | 1 | | | |
| | Vaginal only | 4 | 0.60 | 0.19–1.93 | 0.39 | 0 |
| **Gestational age** | All gestations | 27 | 1 | | | |
| | Term only | 8 | 1.17 | 0.52–2.64 | 0.70 | 0 |
| **Live birth** | All deliveries | 30 | 1 | | | |
| | Live birth only | 5 | 1.41 | 0.55–3.63 | 0.47 | 0 |
| **Low risk** | All women | 28 | 1 | | | |
| | Low-risk pregnancy only | 7 | 0.72 | 0.28–1.84 | 0.49 | 0 |
| **Diagnosis** | Clinical | 30 | 1 | | | |
| | Self-report | 5 | 1.58 | 0.62–4.02 | 0.34 | 0 |
| **Data collection** | Routine | 25 | 1 | | | |
| | Study | 10 | 1.25 | 0.58–2.68 | 0.57 | 0 |
| **Follow-up**\* | Hospital discharge | 20 | 1 | | | |
| | 7 days | 5 | 1.13 | 0.39–3.25 | | |
| | 8–42 days | 9 | 0.87 | 0.38–1.96 | 0.90 | 0 |

\*Length of follow-up was missing for one study. **Abbreviations:** CI, Confidence Interval; OR, odds ratio.

**Maternal peripartum infection.** Incidence of maternal peripartum infection ranged from 0.1%–18.1%, with pooled incidence of 1.9% (95% CI 1.3%–2.8%) (Table 3). The prediction intervals suggest the incidence could be up to 8% in future studies. Pooled incidence in the US and Europe was 1.9%, and in East Asia, it was 2.6%. Other regions contained only one or two studies (Fig 6), and there was no evidence that world region was associated with incidence. In univariate analysis, study extent was strongly associated with incidence. Studies with only low-risk pregnancies or vaginal deliveries also showed some evidence of association, although this was lost after adjusting for study extent (Table 8); many of these studies used either broad or poorly described definitions of infection.

Pooled incidence in seven high-quality studies was 1.1%. The highest incidence of 5.8% was from a single-facility study in China, using Ministry of Health standard diagnosis of genital tract and cesarean section incision infection [30]. All the other estimates extracted ICD-9 or 10 codes for major/other puerperal infection from state or nationally representative hospital data-bases with incidences of 0.2% in Canada and Thailand [74, 78], 0.5% using NIS data [79], 0.8% in all National Health Service (NHS) hospital deliveries in the UK with follow-up to 42 days

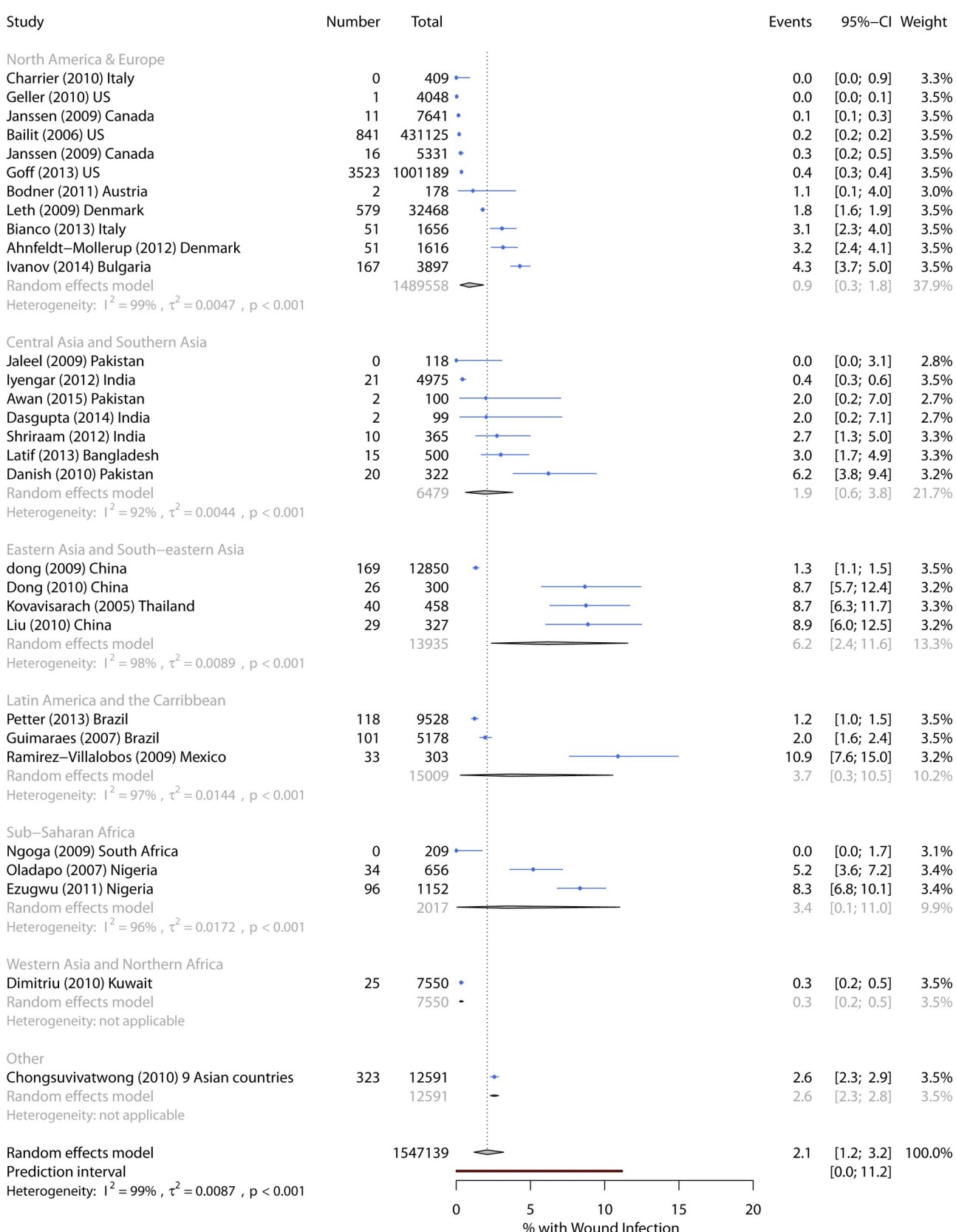

| Study | Number | Total | Events | 95%-CI | Weight |
|---|---|---|---|---|---|
| **North America & Europe** | | | | | |
| Charrier (2010) Italy | 0 | 409 | 0.0 | [0.0; 0.9] | 3.3% |
| Geller (2010) US | 1 | 4048 | 0.0 | [0.0; 0.1] | 3.5% |
| Janssen (2009) Canada | 11 | 7641 | 0.1 | [0.1; 0.3] | 3.5% |
| Bailit (2006) US | 841 | 431125 | 0.2 | [0.2; 0.2] | 3.5% |
| Janssen (2009) Canada | 16 | 5331 | 0.3 | [0.2; 0.5] | 3.5% |
| Goff (2013) US | 3523 | 1001189 | 0.4 | [0.3; 0.4] | 3.5% |
| Bodner (2011) Austria | 2 | 178 | 1.1 | [0.1; 4.0] | 3.0% |
| Leth (2009) Denmark | 579 | 32468 | 1.8 | [1.6; 1.9] | 3.5% |
| Bianco (2013) Italy | 51 | 1656 | 3.1 | [2.3; 4.0] | 3.5% |
| Ahnfeldt−Mollerup (2012) Denmark | 51 | 1616 | 3.2 | [2.4; 4.1] | 3.5% |
| Ivanov (2014) Bulgaria | 167 | 3897 | 4.3 | [3.7; 5.0] | 3.5% |
| Random effects model | | 1489558 | 0.9 | [0.3; 1.8] | 37.9% |
| Heterogeneity: $I^2 = 99\%$, $\tau^2 = 0.0047$, $p < 0.001$ | | | | | |
| **Central Asia and Southern Asia** | | | | | |
| Jaleel (2009) Pakistan | 0 | 118 | 0.0 | [0.0; 3.1] | 2.8% |
| Iyengar (2012) India | 21 | 4975 | 0.4 | [0.3; 0.6] | 3.5% |
| Awan (2015) Pakistan | 2 | 100 | 2.0 | [0.2; 7.0] | 2.7% |
| Dasgupta (2014) India | 2 | 99 | 2.0 | [0.2; 7.1] | 2.7% |
| Shriraam (2012) India | 10 | 365 | 2.7 | [1.3; 5.0] | 3.3% |
| Latif (2013) Bangladesh | 15 | 500 | 3.0 | [1.7; 4.9] | 3.3% |
| Danish (2010) Pakistan | 20 | 322 | 6.2 | [3.8; 9.4] | 3.2% |
| Random effects model | | 6479 | 1.9 | [0.6; 3.8] | 21.7% |
| Heterogeneity: $I^2 = 92\%$, $\tau^2 = 0.0044$, $p < 0.001$ | | | | | |
| **Eastern Asia and South−eastern Asia** | | | | | |
| dong (2009) China | 169 | 12850 | 1.3 | [1.1; 1.5] | 3.5% |
| Dong (2010) China | 26 | 300 | 8.7 | [5.7; 12.4] | 3.2% |
| Kovavisarach (2005) Thailand | 40 | 458 | 8.7 | [6.3; 11.7] | 3.3% |
| Liu (2010) China | 29 | 327 | 8.9 | [6.0; 12.5] | 3.2% |
| Random effects model | | 13935 | 6.2 | [2.4; 11.6] | 13.3% |
| Heterogeneity: $I^2 = 98\%$, $\tau^2 = 0.0089$, $p < 0.001$ | | | | | |
| **Latin America and the Carribbean** | | | | | |
| Petter (2013) Brazil | 118 | 9528 | 1.2 | [1.0; 1.5] | 3.5% |
| Guimaraes (2007) Brazil | 101 | 5178 | 2.0 | [1.6; 2.4] | 3.5% |
| Ramirez−Villalobos (2009) Mexico | 33 | 303 | 10.9 | [7.6; 15.0] | 3.2% |
| Random effects model | | 15009 | 3.7 | [0.3; 10.5] | 10.2% |
| Heterogeneity: $I^2 = 97\%$, $\tau^2 = 0.0144$, $p < 0.001$ | | | | | |
| **Sub−Saharan Africa** | | | | | |
| Ngoga (2009) South Africa | 0 | 209 | 0.0 | [0.0; 1.7] | 3.1% |
| Oladapo (2007) Nigeria | 34 | 656 | 5.2 | [3.6; 7.2] | 3.4% |
| Ezugwu (2011) Nigeria | 96 | 1152 | 8.3 | [6.8; 10.1] | 3.4% |
| Random effects model | | 2017 | 3.4 | [0.1; 11.0] | 9.9% |
| Heterogeneity: $I^2 = 96\%$, $\tau^2 = 0.0172$, $p < 0.001$ | | | | | |
| **Western Asia and Northern Africa** | | | | | |
| Dimitriu (2010) Kuwait | 25 | 7550 | 0.3 | [0.2; 0.5] | 3.5% |
| Random effects model | | 7550 | 0.3 | [0.2; 0.5] | 3.5% |
| Heterogeneity: not applicable | | | | | |
| **Other** | | | | | |
| Chongsuvivatwong (2010) 9 Asian countries | 323 | 12591 | 2.6 | [2.3; 2.9] | 3.5% |
| Random effects model | | 12591 | 2.6 | [2.3; 2.8] | 3.5% |
| Heterogeneity: not applicable | | | | | |
| **Random effects model** | | 1547139 | 2.1 | [1.2; 3.2] | 100.0% |
| Prediction interval | | | | [0.0; 11.2] | |
| Heterogeneity: $I^2 = 99\%$, $\tau^2 = 0.0087$, $p < 0.001$ | | | | | |

0  5  10  15  20
% with Wound Infection

**Fig 4. Forest plot of wound infection incidence by world region.** CI, Confidence Interval.

**Table 6. Wound metaregression.**

| Factor | | No. of Studies | OR | 95% CI | *p*-Value | R2 (%) | Adj. OR | 95% CI |
|---|---|---|---|---|---|---|---|---|
| | | | | | | | $R^2$ = 43.78% | |
| Region | North America and Europe | 11 | 1 | | 0.02 | 25.2 | 1 | |
| | Central Asia and South Asia | 7 | 3 | 0.83–10.82 | | | 1.84 | 0.48–7.12 |
| | East Asia and Southeast Asia | 4 | 9.1 | 2.11–39.20 | | | 3.85 | 0.89–16.72 |
| | Latin America and the Caribbean | 3 | 4.85 | 0.96–24.52 | | | 2.06 | 0.42–10.06 |
| | Sub-Saharan Africa | 3 | 5.98 | 1.03–34.69 | | | 2.75 | 0.50–15.22 |
| | Western Asia and Northern Africa | 1 | 0.52 | | | | 0.22 | 0.02–2.37 |
| Study extent | Single site | 22 | 1 | | 0.002 | 37.9 | | |
| | 2+ sites | 2 | 0.11 | 0.02–0.80 | | | 0.13 | 0.02–0.94 |
| | State | 4 | 0.13 | 0.04–0.46 | | | 0.24 | 0.05–1.04 |
| | National | 1 | 0.13 | 0.01–1.30 | | | 0.23 | 0.02–2.44 |
| Number of foetuses | All pregnancies | 21 | 1 | | | | | |
| | Singleton only | 8 | 1.95 | 0.56–6.75 | 0.29 | 3.5 | | |
| Delivery mode | All deliveries | 24 | 1 | | | | | |
| | Vaginal only | 5 | 4.64 | 1.21–17.76 | 0.02 | 17.8 | | |
| Gestational age | All gestations | 24 | 1 | | | | | |
| | Term only | 5 | 0.85 | 0.18–4.08 | 0.84 | 0 | | |
| Live birth | All deliveries | 26 | 1 | | | | | |
| | Live birth only | 3 | 1.31 | 0.22–7.76 | 0.76 | 0 | | |
| Low risk | All women | 21 | 1 | | | | | |
| | Low-risk pregnancy only | 8 | 0.60 | 0.17–2.14 | 0.43 | 0 | | |
| Diagnosis | Clinical | 25 | 1 | | | | | |
| | Self-report | 4 | 1.58 | 0.62–4.02 | 0.33 | 0 | | |
| Data collection | Routine | 16 | 1 | | | | | |
| | Study | 8 | 2.99 | 0.87–10.25 | | | | |
| | Unclear | 5 | 1.92 | 0.40–9.19 | 0.21 | 5.9 | | |
| Follow-up* | Discharge | 17 | 1 | | | | | |
| | Day 7 | 2 | 3.57 | 0.42–30.25 | | | | |
| | 8–42 days | 8 | 1.26 | 0.38–4.22 | 0.50 | 0 | | |

*Length of follow-up was missing from two studies. **Abbreviations:** Adj., adjusted; CI, Confidence Interval; OR, odds ratio.

[80], and 0.9% using birth certificate data in California [81]. One large US study also included chorioamnionitis and reported 2.0% of women with infection [82].

## Discussion

We systematically reviewed the incidence of maternal peripartum infection and identified 111 studies from 46 countries, representing all world regions, from among 31,528 potential studies. Pooled infection incidence in high-quality studies was 3.9% (95% CI 1.8%–6.8%) for chorioamnionitis, 1.6% (95% CI 0.9%–2.5%) for endometritis, 1.2% (95% CI 1.0%–1.5%) for wound infection (one study), and 1.1% (95% CI 0.3%–2.4%) for maternal peripartum infection. Pooled incidence of sepsis was 0.05% (95% CI 0.03%–0.07%). Studies of composite outcomes had, on average, a lower incidence than obtained by summing other infection outcomes (1.1% versus 6.7%), probably because they rarely included chorioamnionitis (3.9%) but also because coinfections can occur.

Comparing our results to other global estimates is complicated by the different definitions used. The recent 2017 GBD global incidence of maternal infection of 12.1 million women [83]

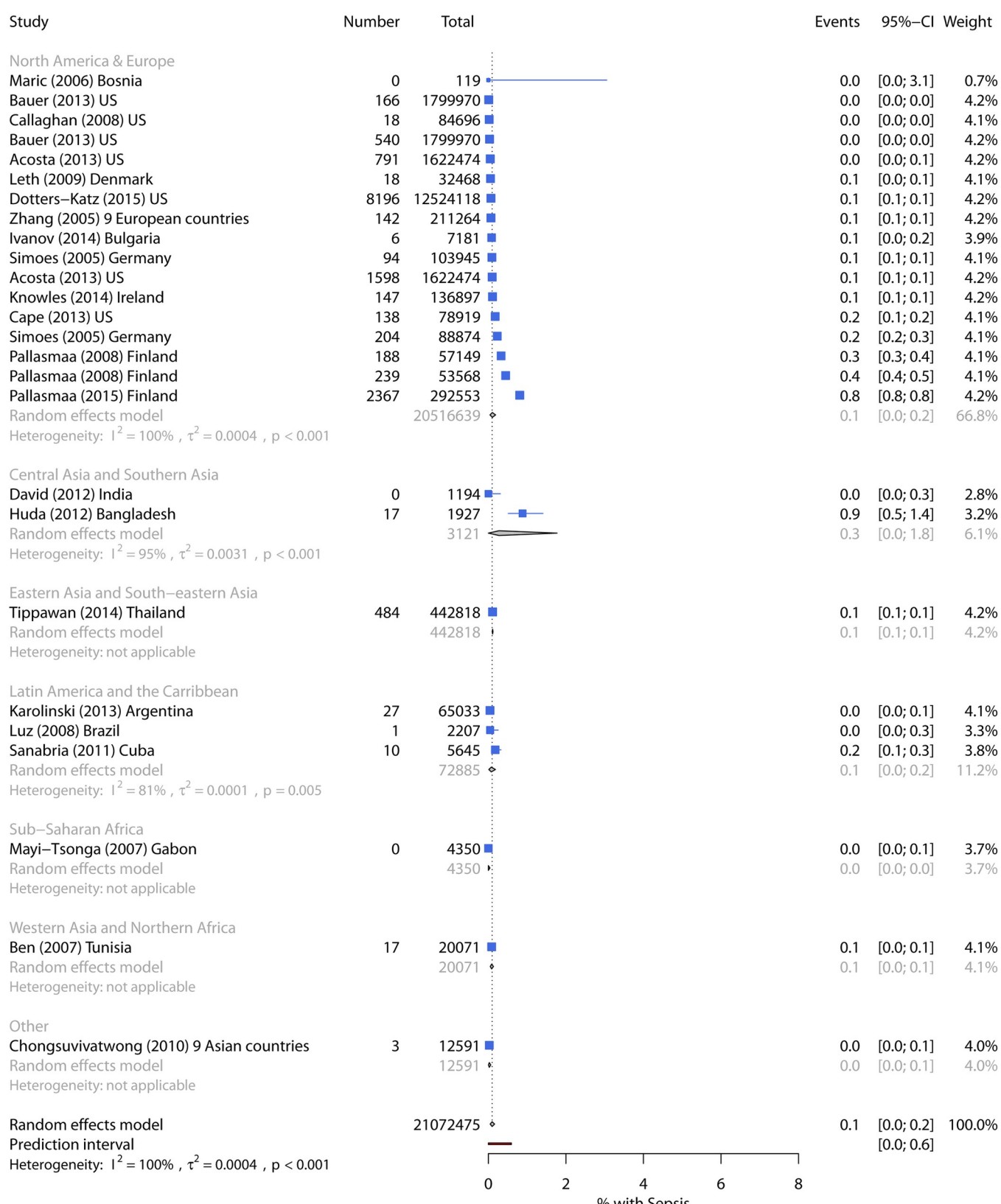

**Fig 5. Forest plot of sepsis incidence by world region.** CI, Confidence Interval.

**Table 7. Sepsis metaregression.**

| Factor | | No. of Studies | OR | 95% CI | p-Value | R² (%) |
|---|---|---|---|---|---|---|
| **Severity** | SIRS* | 13 | 1 | | | |
| | Severe sepsis | 5 | 0.32 | 0.08–1.35 | | |
| | Septicaemia/peritonitis | 7 | 0.52 | 0.15–1.78 | 0.25 | 2.6 |
| **Region** | North America and Europe | 16 | 1 | | | |
| | Central Asia and South Asia | 3 | 11.00 | 2.25–53.75 | | |
| | East Asia and Southeast Asia | 1 | 1.23 | 0.12–12.50 | | |
| | Latin America and the Caribbean | 3 | 0.83 | 0.18–3.84 | | |
| | Sub-Saharan Africa | 1 | 0.13 | 0.004–4.79 | | |
| | West Asia and North Africa | 1 | 0.96 | 0.09–10.15 | 0.06 | 25.1 |
| **Study extent** | Single site | 8 | 1 | | | |
| | 2+ sites | 2 | 6.84 | 0.83–56.64 | | |
| | Network | 2 | 2.06 | 0.25–17.12 | | |
| | State | 6 | 0.92 | 0.21–4.08 | | |
| | National | 7 | 0.83 | 0.20–3.50 | 0.32 | 2.5 |
| **Number of foetuses** | All deliveries | 23 | 1 | | | |
| | Singleton only | 2 | 6.64 | 1.11–39.63 | 0.04 | 13.5 |
| **Delivery mode** | All deliveries | 23 | 1 | | | |
| | Vaginal only | 2 | 1.24 | 0.08–19.58 | 0.88 | 0 |
| **Gestational age** | All gestations | 25 | – | | | |
| | Term only | 0 | | | | |
| **Live birth** | All deliveries | 24 | 1 | | | |
| | Live birth only | 1 | 0.37 | 0.02–5.54 | 0.47 | 0 |
| **Low risk** | All women | 24 | 1 | | | |
| | Low-risk pregnancy only | 1 | 0.42 | 0.01–14.91 | 0.64 | 0 |
| **Diagnosis** | Clinical | 25 | | | | |
| | Self-report | 0 | | | | |
| **Data collection** | Routine | 24 | 1 | | | |
| | Study | 1 | 2.99 | 0.87–10.25 | | |
| | Unclear | 1 | 1.92 | 0.40–9.19 | 0.21 | 5.9 |
| **Follow-up*** | Discharge/day 7 | 13 | 1 | | | |
| | Day 8–42 | 10 | 3.57 | 1.55–8.22 | 0.003 | 27.2 |

*Length of follow-up was missing for two studies. **Abbreviations:** CI, Confidence Interval; OR, odds ratio; SIRS, systemic inflammatory response syndrome.

translates to an estimated 8.2% of live births [84] but includes mastitis, so it is not comparable with ours. Dolea and Stein's older figure of 4% for puerperal sepsis [6] excludes surgical site infection (SSI) but includes urinary tract infection. Our average estimates of endometritis, maternal peripartum infection, and sepsis are all substantially lower, which may reflect our exclusion of urinary tract infection or a reduction in infection since 2000. Our identification of source estimates is vastly more comprehensive than either GBD or Dolea and Stein, and we do not rely on modelling. A recently published review of infection following cesarean section in sub-Saharan Africa reports an SSI rate of 15.6% that, at their reported cesarean section rate of 12.4%, corresponds to 1.9% for the total population of women giving birth [85]. This is a little lower than the average incidence (3.4%) in our three fairly small, poor-quality African studies but does not include perineal wound infection and does lie within our prediction interval.

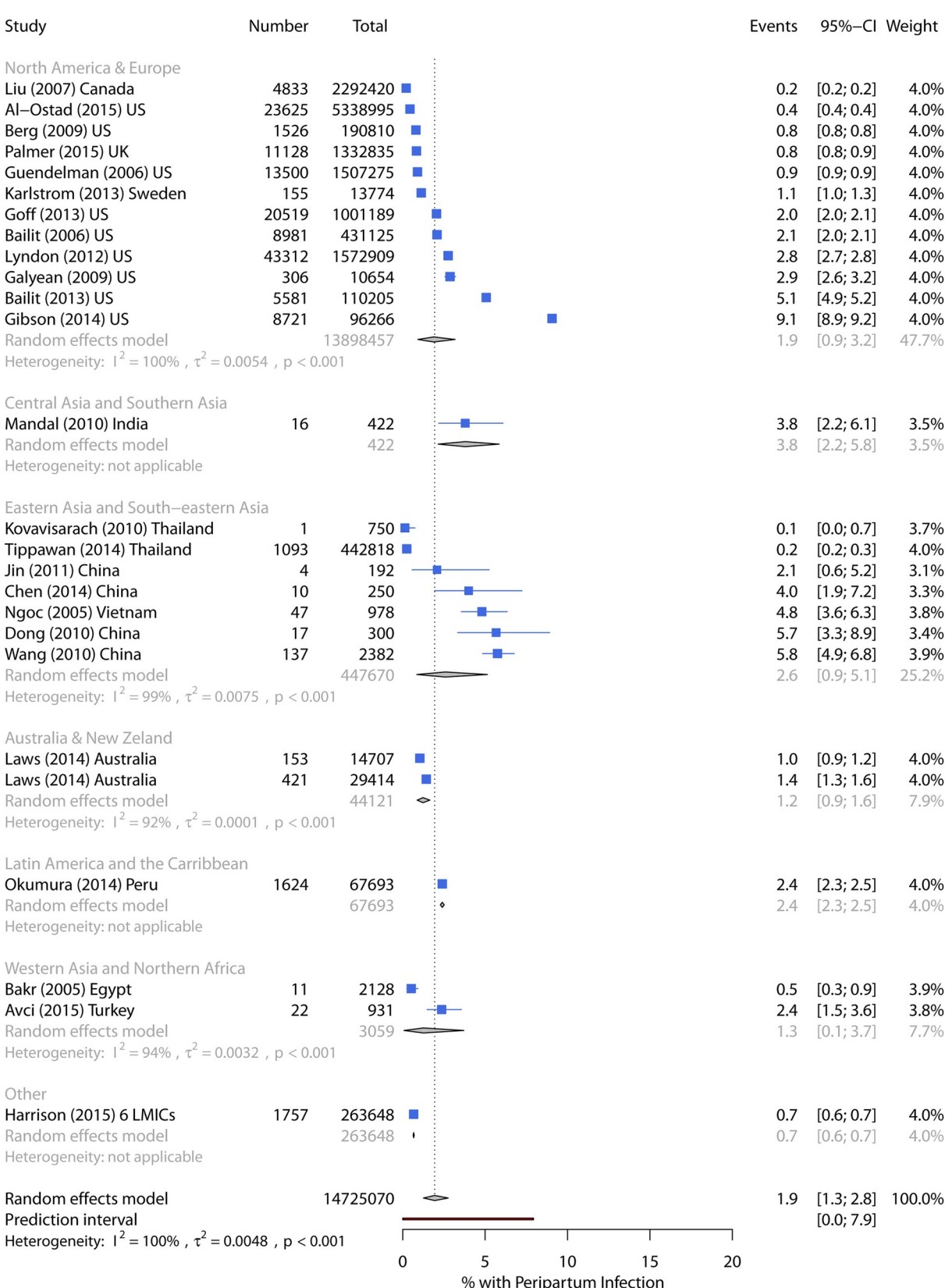

| Study | Number | Total | | Events | 95%–CI | Weight |
|---|---|---|---|---|---|---|
| **North America & Europe** | | | | | | |
| Liu (2007) Canada | 4833 | 2292420 | | 0.2 | [0.2; 0.2] | 4.0% |
| Al–Ostad (2015) US | 23625 | 5338995 | | 0.4 | [0.4; 0.4] | 4.0% |
| Berg (2009) US | 1526 | 190810 | | 0.8 | [0.8; 0.8] | 4.0% |
| Palmer (2015) UK | 11128 | 1332835 | | 0.8 | [0.8; 0.9] | 4.0% |
| Guendelman (2006) US | 13500 | 1507275 | | 0.9 | [0.9; 0.9] | 4.0% |
| Karlstrom (2013) Sweden | 155 | 13774 | | 1.1 | [1.0; 1.3] | 4.0% |
| Goff (2013) US | 20519 | 1001189 | | 2.0 | [2.0; 2.1] | 4.0% |
| Bailit (2006) US | 8981 | 431125 | | 2.1 | [2.0; 2.1] | 4.0% |
| Lyndon (2012) US | 43312 | 1572909 | | 2.8 | [2.7; 2.8] | 4.0% |
| Galyean (2009) US | 306 | 10654 | | 2.9 | [2.6; 3.2] | 4.0% |
| Bailit (2013) US | 5581 | 110205 | | 5.1 | [4.9; 5.2] | 4.0% |
| Gibson (2014) US | 8721 | 96266 | | 9.1 | [8.9; 9.2] | 4.0% |
| Random effects model | | 13898457 | | 1.9 | [0.9; 3.2] | 47.7% |
| Heterogeneity: $I^2 = 100\%$ , $\tau^2 = 0.0054$ , $p < 0.001$ | | | | | | |
| **Central Asia and Southern Asia** | | | | | | |
| Mandal (2010) India | 16 | 422 | | 3.8 | [2.2; 6.1] | 3.5% |
| Random effects model | | 422 | | 3.8 | [2.2; 5.8] | 3.5% |
| Heterogeneity: not applicable | | | | | | |
| **Eastern Asia and South–eastern Asia** | | | | | | |
| Kovavisarach (2010) Thailand | 1 | 750 | | 0.1 | [0.0; 0.7] | 3.7% |
| Tippawan (2014) Thailand | 1093 | 442818 | | 0.2 | [0.2; 0.3] | 4.0% |
| Jin (2011) China | 4 | 192 | | 2.1 | [0.6; 5.2] | 3.1% |
| Chen (2014) China | 10 | 250 | | 4.0 | [1.9; 7.2] | 3.3% |
| Ngoc (2005) Vietnam | 47 | 978 | | 4.8 | [3.6; 6.3] | 3.8% |
| Dong (2010) China | 17 | 300 | | 5.7 | [3.3; 8.9] | 3.4% |
| Wang (2010) China | 137 | 2382 | | 5.8 | [4.9; 6.8] | 3.9% |
| Random effects model | | 447670 | | 2.6 | [0.9; 5.1] | 25.2% |
| Heterogeneity: $I^2 = 99\%$ , $\tau^2 = 0.0075$ , $p < 0.001$ | | | | | | |
| **Australia & New Zeland** | | | | | | |
| Laws (2014) Australia | 153 | 14707 | | 1.0 | [0.9; 1.2] | 4.0% |
| Laws (2014) Australia | 421 | 29414 | | 1.4 | [1.3; 1.6] | 4.0% |
| Random effects model | | 44121 | | 1.2 | [0.9; 1.6] | 7.9% |
| Heterogeneity: $I^2 = 92\%$ , $\tau^2 = 0.0001$ , $p < 0.001$ | | | | | | |
| **Latin America and the Carribbean** | | | | | | |
| Okumura (2014) Peru | 1624 | 67693 | | 2.4 | [2.3; 2.5] | 4.0% |
| Random effects model | | 67693 | | 2.4 | [2.3; 2.5] | 4.0% |
| Heterogeneity: not applicable | | | | | | |
| **Western Asia and Northern Africa** | | | | | | |
| Bakr (2005) Egypt | 11 | 2128 | | 0.5 | [0.3; 0.9] | 3.9% |
| Avci (2015) Turkey | 22 | 931 | | 2.4 | [1.5; 3.6] | 3.8% |
| Random effects model | | 3059 | | 1.3 | [0.1; 3.7] | 7.7% |
| Heterogeneity: $I^2 = 94\%$ , $\tau^2 = 0.0032$ , $p < 0.001$ | | | | | | |
| **Other** | | | | | | |
| Harrison (2015) 6 LMICs | 1757 | 263648 | | 0.7 | [0.6; 0.7] | 4.0% |
| Random effects model | | 263648 | | 0.7 | [0.6; 0.7] | 4.0% |
| Heterogeneity: not applicable | | | | | | |
| Random effects model | | 14725070 | | 1.9 | [1.3; 2.8] | 100.0% |
| Prediction interval | | | | | [0.0; 7.9] | |
| Heterogeneity: $I^2 = 100\%$ , $\tau^2 = 0.0048$ , $p < 0.001$ | | | | | | |

0   5   10   15   20
% with Peripartum Infection

**Fig 6. Forest plot of maternal peripartum infection incidence by world region.** CI, Confidence Interval; LMICs, low- and middle-income countries.

**Table 8. Maternal peripartum infection metaregression.**

| Factor | | No. of Studies | OR | 95% CI | *p*-Value | R² (%) | Adj. OR | 95% CI |
|---|---|---|---|---|---|---|---|---|
| | | | | | | | R² = 35.7% | |
| **Region** | North America and Europe | 12 | 1 | | | | | |
| | Central Asia and South Asia | 1 | 2.63 | 0.24–28.80 | | | | |
| | East Asia and Southeast Asia | 7 | 1.37 | 0.45–4.16 | | | | |
| | Australia and New Zealand | 2 | 0.82 | 0.15–4.61 | | | | |
| | Latin America and the Caribbean | 1 | 1.64 | 0.16–17.05 | | | | |
| | West Asia and North Africa | 2 | 0.76 | 0.13–4.38 | 0.93 | 0 | | |
| **Study extent** | Single site | 9 | 1 | | | | 1 | |
| | 2+ sites | 5 | 1.22 | 0.47–3.17 | | | 1.32 | 0.50–3.48 |
| | Network | 1 | 2.20 | 0.38–12.80 | | | 1.54 | 0.24–9.87 |
| | State | 3 | 0.72 | 0.23–2.24 | | | 0.88 | 0.27–2.85 |
| | National | 7 | 0.26 | 0.10–0.61 | 0.005 | 35.6 | 0.29 | 0.12–0.70 |
| **Number of foetuses** | All deliveries | 14 | 1 | | | | | |
| | Singleton only | 11 | 1.66 | 0.71–3.87 | 0.24 | 0.7 | | |
| **Delivery mode** | All deliveries | 22 | 1 | | | | | |
| | Vaginal only | 3 | 3.83 | 1.16–12.67 | 0.03 | 14.3 | | |
| **Gestational age** | All gestations | 17 | 1 | | | | | |
| | Term only | 8 | 0.89 | 0.36–2.23 | 0.81 | 0 | | |
| **Live birth** | All deliveries | 20 | 1 | | | | | |
| | Liver birth only | 5 | 1.61 | 0.57–4.59 | 0.37 | 0 | | |
| **Low risk** | All women | 19 | 1 | | | | 1 | |
| | Low-risk pregnancy only | 6 | 2.34 | 0.90–6.04 | 0.08 | 7.3 | 1.74 | 0.71–4.27 |
| **Diagnosis** | Clinical | 24 | – | | | | | |
| | Unclear | 1 | | | | | | |
| **Data collection** | Routine | 18 | 1 | | | | | |
| | Study | 3 | 2.67 | 0.71–10.10 | | | | |
| | Unclear | 4 | 0.74 | 0.22–2.52 | 0.28 | 1.5 | | |
| **Follow-up** | Discharge | 20 | 1 | | | | | |
| | Until day 42 | 5 | 1.17 | 0.40–3.41 | 0.77 | 0 | | |

**Abbreviations:** Adj., adjusted; CI, Confidence Interval; OR, odds ratio.

## Limitations of included studies

The quality of many studies was poor, with potential for bias. Measurement bias was possible in 63% of studies, primarily because the infection was not defined, or the definition used was too broad and risked overestimating incidence. This explains part of the between-study heterogeneity observed. Attrition was minimal because most studies were cross-sectional or had short follow-up periods. There was potential selection bias in nearly one-third of studies: most trials did not describe initial selection methods, and pair-matched studies produced nonrandom control groups; however, it is unclear whether and how this might have affected infection incidence. Restricting the results to high-quality studies made little difference to the pooled estimates for chorioamnionitis or endometritis but produced lower pooled incidence for the other outcomes, although with similar prediction intervals. This lower incidence may be an underestimate of infection because some high-quality studies had narrower outcome definitions than the standards. In addition, only one lower-middle–income and four upper-middle–income countries contributed to high-quality estimates, reducing their generalisability to LMICs.

We explored and quantified the importance of world region and study characteristics on infection risk using metaregression to explain heterogeneity and better compare study estimates. Unfortunately, our analyses were limited by data sparsity. Beyond North America and Europe, data were scarce, especially from sub-Saharan Africa and Western Asia and North Africa. We found some evidence for increased wound infection outside North America and Europe but saw a mixed picture for endometritis, with surprisingly low incidence in East and Southeast Asia. In common with other studies, we found a higher incidence of SSI in LMICs, which could reflect differences in surgical and infection control practices [86]. However, studies outside North America and Europe were also more likely to be at single facilities, use self-reported symptoms, and collect data specifically for the study—all features that relate to higher incidence.

For chorioamnionitis, wound infection, and maternal peripartum infection, there was evidence that study extent was associated with infection risk. Pooled incidence was up to five times higher in single-facility studies compared to estimates using nationally representative databases, although the association was less clear with state-level studies. Large databases relying on routine medical records risk underestimating incidence because of missing or misclassified data. Conversely, studies at single tertiary-level hospitals may represent higher risk populations, especially in LMICs with low facility delivery rates, producing overestimates of population-level incidence. We excluded studies of high-risk women from this review but chose to retain single-facility studies and regress the effect of study extent on infection because omitting single facilities would lead to extensive loss of data, especially from LMICs.

Longer follow-up (risk) period was unsurprisingly associated with higher sepsis incidence, and a similar trend was observed with wound infection but lacked statistical evidence. This supports the findings of one included study in which the majority of infections occurred after hospital discharge [87]. Unfortunately, the majority of studies only collected data during hospital admission and may therefore have missed many cases.

Expected low-risk groups, including live, term, singleton, and vaginal births, did not have a lower infection risk compared to studies of all deliveries. This was surprising, but because the majority of deliveries, even in population-level studies, are also low-risk, it is difficult to show evidence of a difference. Occasionally, there was evidence of higher infection incidence in the studies of low-risk groups, but numbers were often small, and results were confounded by other study design factors.

### Strengths and limitations of review

This review's strengths include the very extensive search conducted and the inclusion of articles in all languages identified. However, studies published after June 2016 have not contributed to the findings. Our review adopted the 2015 WHO definition of maternal peripartum infections and used international standard definitions among its quality criteria. It could be criticised for not restricting included studies to those meeting the full WHO definition, including the specified time period from onset of labour until 42 days postpartum. However, it is telling that none of the studies measured this exact outcome, and very few of those investigating postpartum infection continued until 42 days.

The review reported infection outcomes as an incident risk. This assumes all women were at risk (i.e., free of the infections under consideration) at the start of follow-up: onset of labour or immediately postpartum. However, some studies were unable or did not seek to exclude women with existing infections, potentially overestimating the incidence. Some studies only assessed or interviewed women at one time point after delivery; however, follow-up periods were short, so the chance of missing infections is small. We excluded studies that only assessed

high-risk subgroups of women; however, we did not limit our review to population-level studies, potentially overestimating infection incidence, as discussed above. Conversely, we did include groups of low-risk women, and so our pooled estimates may be an underestimate.

There are arguments against pooling estimates in the presence of extensive heterogeneity. Although $I^2$ was very high, this is driven by the substantial number of large, precise studies [56]. $Tau^2$ is a more relevant measure of heterogeneity in this case, and values were small. Moreover, we believe that within our outcome groups, each study was attempting to measure the same outcome, and therefore, the average estimates remain useful, although they should be treated cautiously and not overinterpreted as measures of global incidence.

## Conclusion

To our knowledge, this is the first global systematic review of maternal peripartum infection incidence. It demonstrates that infection is an important complication of childbirth. Moreover, we found that a large proportion of these infections occurred in labour, with implications for the baby and the mother. Postpartum infection incidence appears lower than modelled global estimates, although the difference in definition limits comparability, and the proportion of women affected is still considerable. At a time of growing concern about antimicrobial resistance, these findings highlight the importance for clinicians and policymakers to focus efforts on improved infection prevention practices to reduce this preventable cause of maternal morbidity. Our study provides useful estimates to guide sample-size calculations for future intervention research. However, we also highlight the paucity of data from LMICs and the heterogeneity in study designs, quality, and infection definitions. Better-quality research, using standard definitions and follow-up after hospital discharge, is required to improve comparability between different study settings and to demonstrate the influence of risk factors and protective interventions.

## Supporting information

**S1 Checklist. PRISMA checklist.** PRISMA, Preferred Reporting Items for Systematic Reviews and Meta-Analyses.
(DOC)

**S1 Text. Search strategy.**
(DOCX)

**S2 Text. Data extraction form.**
(DOCX)

**S1 Table. ICD codes for infection outcomes.**
(DOCX)

**S2 Table. Studies of chorioamnionitis.**
(DOCX)

**S3 Table. Studies of endometritis.**
(DOCX)

**S4 Table. Studies of wound infection.**
(DOCX)

**S5 Table. Studies of sepsis.**
(DOCX)

**S6 Table. Studies of maternal peripartum infection.**
(DOCX)

**S7 Table. Quality of 111 included studies.**
(DOCX)

**S1 Fig. Forest plot of sepsis incidence by severity.**
(TIF)

# Acknowledgments

Jane Falconer (London School of Hygiene and Tropical Medicine [LSHTM] library) and Giorgia Gon (LSHTM) advised on the search strategy, and staff at WHO and the LSHTM library assisted with locating full texts of articles. Jeff Brunton and the team at Institute of Education gave technical support in using Eppi-Reviewer software. Sylvia Marinova (LSHTM), Petra Klepac (LSHTM), Veronique Filippi (LSHTM), Vladimir Gordeev (LSHTM), Bagheri Nejad (WHO), and Chaza Akik screened and translated non-English–language papers. Lale Say, as principal investigator of the WHO Maternal Morbidity Working Group, led the conceptualisation of the whole maternal morbidity project that acted as the incentive for this review and provided a generic protocol to work from and a platform to discuss progress. She also acted as guarantor of all scientific activities and provided critical feedback on the article, together with other members of the group: Benedetta Allegranzi, Mercedes Bonet Semenas, Jennifer Cresswell, and Ann-Beth Moller. Wendy Graham and The Soapbox Collaborative provided the time and encouragement to start and complete the work.

# Author Contributions

**Conceptualization:** Susannah L. Woodd, Doris Chou.

**Data curation:** Susannah L. Woodd.

**Formal analysis:** Susannah L. Woodd, Clara Calvert, Andrea M. Rehman.

**Funding acquisition:** Susannah L. Woodd, Doris Chou.

**Investigation:** Susannah L. Woodd, Ana Montoya, Maria Barreix, Li Pi.

**Methodology:** Susannah L. Woodd, Ana Montoya, Clara Calvert, Andrea M. Rehman, Oona M. R. Campbell.

**Project administration:** Susannah L. Woodd.

**Supervision:** Doris Chou, Oona M. R. Campbell.

**Visualization:** Susannah L. Woodd, Clara Calvert.

**Writing – original draft:** Susannah L. Woodd, Oona M. R. Campbell.

**Writing – review & editing:** Susannah L. Woodd, Ana Montoya, Maria Barreix, Li Pi, Clara Calvert, Andrea M. Rehman, Doris Chou, Oona M. R. Campbell.

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
