## [Decision Letter · Decision Letter 0]

4 Sep 2019

Dear Dr. Woodd,

Thank you very much for submitting your manuscript "Incidence of maternal peripartum infection: A systematic literature review and meta-analysis" (PMEDICINE-D-19-02904) for consideration at PLOS Medicine. 

[LINK]

In light of these reviews, I am afraid that we will not be able to accept the manuscript for publication in the journal in its current form, but we would like to consider a revised version that addresses the reviewers' and editors' comments. Obviously we cannot make any decision about publication until we have seen the revised manuscript and your response, and we plan to seek re-review by one or more of the reviewers. 

We expect to receive your revised manuscript by Sep 25 2019 11:59PM. Please email us (plosmedicine@plos.org) if you have any questions or concerns.

We look forward to receiving your revised manuscript. 

Sincerely,

Louise Gaynor, MBBS PhD

Associate Editor 

PLOS Medicine

plosmedicine.org

General comments

Please add a space before reference brackets.

Please add full stops after the reference brackets at the end of the sentences.

Title

Please omit the word ‘literature’ 

Abstract

Please report your abstract according to PRISMA for abstracts, following the PLOS Medicine abstract structure (Background, Methods and Findings, Conclusions) http://www.plosmedicine.org/article/info:doi/10.1371/journal.pmed.1001419

Abstract - Background

Provide the context of why the study is important. The final sentence should clearly state the study question.

Please avoid assertions of primacy. Please add ‘to our knowledge’ to your final sentence. 

Abstract - Methods & Findings

Please provide the data sources and types of study designs included (including brief demographic details). 

Please ensure that all numbers presented in the abstract are present and identical to numbers presented in the main manuscript text.

In the last sentence of the Abstract Methods and Findings section, please describe the main limitation(s) of the study's methodology.

Please capitalise PROSPERO

Please specify that numbers in brackets are 95% CI 

Abstract - Conclusions 

Please begin your conclusion with ‘In this study, we observed …’ or similar 

Author Summary 

Introduction

Please specify that numbers in brackets are 95% CI 

Please define puerperium

Please explain the need for and potential importance of your study. 

Indicate whether your study is novel and how you determined that. 

Please conclude the Introduction with a clear description of the study question or hypothesis.

Methods 

Please update your search to the present time. We require that SRs are updated to within roughly 6 months of the expected publication date.

Please capitalise PROSPERO

Please report your SR/MA according to the appropriate study design provided at the EQUATOR site. http://www.equator-network.org/?post_type=eq_guidelines&eq_guidelines_study_design=systematic-reviews-and-meta-analyses&eq_guidelines_clinical_specialty=0&eq_guidelines_report_section=0&s=+

Please include sufficient text excerpted from the manuscript to explain how you accomplished all applicable checklist items. When completing the checklist, please use section and paragraph numbers from within each section, rather than page numbers. 

Please refer to the attached PRISMA checklist early in your methods section. 

Please provide names of all nine databases (as mentioned in your abstract) 

Please avoid terms like ‘general’ 

Please define JHPIEGO

Results 

When a p value is given, please specify the statistical test used to determine it.

Line 225 - please clarify what is meant by ‘Attrition or missing data of >15% was described in 13%‘

Line 252 - please define NIS

Line 313 - please correct ‘severe sepsis with or organ dysfunction’

Line 315 - please define NHDS

Discussion 

Please present and organize the Discussion as follows: a short, clear summary of the article's findings; what the study adds to existing research and where and why the results may differ from previous research; strengths and limitations of the study; implications and next steps for research, clinical practice, and/or public policy; one-paragraph conclusion.

At line 416, please substitute "very" for "extremely".

Conclusion

Please avoid assertions of primacy. Please add ‘to our knowledge’.

References

Please ensure that journal titles are capitalised properly.

To references 24, 45 & 66, please full access details (journal number, page numbers or URL, as available).

Comments from the reviewers:

Reviewer #1: This is a worthy paper and, although i'm not an epidemiologist, I have no scientific concersn with the methodology. 

My only worry is whether it is of sufficient general interest to justify publication is such a high impact general journal as PLOS medicine. As the authors note, previous estimates of perinatal infection have been around 3%. This review confirms that figure to be broadly correct, although the estimate is very uncertain because of the different ways in which infection has been identified and classified. And there you have it.

IMHO worthy of publication somehwere but not suffciently high priority for PLOS Medicine. 

Reviewer #2: See attachment

Michael Dewey

Reviewer #3: ¬¬¬Review of manuscript P-Medicine 19-02904

Thank you for the opportunity to review this manuscript. This is a systematic literature review and meta-analysis on incidence of maternal peripartum infection. The study used random-effects models to obtain pooled estimates for each outcome. Incidence of peripartum infection risk appear to be lower than global estimates. The study used the WHO definition for peripartum infection of the genital tract or surrounding tissues from onset of membrane ruptures to 42 days postpartum. In general, the study is of general interest and of high quality. 

General comments

1. Not clear how the outcome "Maternal peripartum infection" is defined. Is it any of the infections studied? An overall measure? Then why is the incidence lower as compared to chorioamnionitis?

2. Infection incidences varies with mode of delivery and the rate of caesarean delivery differ between countries, regions and clinics. Hence, wound infection would benefit from being reported for caesarean deliveries separately.

3. Why were infections in the urinary tract excluded? One could argue that they would be included in the genital tract.

Specific comments

1. Would it be possible to provide a table on the ICD-codes used in various studies for assessment of the subgroups of peripartum infections?

2. Table 3 appear to be truncated (ends with Liu).

3. Could the study also assess instrumental deliveries (vacuum and forceps)?

Reviewer #4: Review of PMEDICINE-D-19-0290

The authors report a meta-analysis on the incidence of peripartum infection. The subject is important clinically.

The reported incidence of chorioamnionitis, endometritis, wound infection and sepsis are consistent with contemporary literature and useful for counseling and quality assessment/ improvement among centers.

The high heterogeneity in definition of infections is concerning and questions the utility of pooling the data together. I agree with the conclusion that "Better quality research, using standard definitions, is required ….".

Specific points:

1. Please justify mixing clinical trials with observational studies. 

2. I agree with reporting the random effects models when definitions of infection varied widely, but can the authors justify pooling when the heterogeneity is > 99%?

3. The authors make a good argument in the discussion for not limiting the analyses to studies meeting the WHO criteria for infection, but a sensitivity analyses doing this and reporting the findings could be helpful.

[LINK]

---

## [Decision Letter · Decision Letter 1]

23 Oct 2019

Dear Dr. Woodd,

Thank you very much for re-submitting your manuscript "Incidence of maternal peripartum infection: A systematic review and meta-analysis" (PMEDICINE-D-19-02904R1) for review by PLOS Medicine.

I have discussed the paper with my colleagues and the academic editor and it was also seen again by reviewers. I am pleased to say that provided the remaining editorial and production issues are dealt with we are planning to accept the paper for publication in the journal.

[LINK]

We look forward to receiving the revised manuscript by Oct 30 2019 11:59PM. 

Sincerely,

Clare Stone

Managing Editor

PLOS Medicine

plosmedicine.org

Requests from Editors:

Data needs to be deposited / URL provided and noting that an author cannot be a point of contact. We are unable to publish unless / until this is done. PLOS Medicine requires that the de-identified data underlying the specific results in a published article be made available, without restrictions on access, in a public repository or as Supporting Information at the time of article publication, provided it is legal and ethical to do so. Please see the policy at 

http://journals.plos.org/plosmedicine/s/data-availability

and FAQs at 

http://journals.plos.org/plosmedicine/s/data-availability#loc-faqs-for-data-policy

For the limitations (last sentence of the methods and findings section of the abstract), please do start this sentence in a more explicit way “ limitations of this study are….”

Ref 6 needs completing, please

Comments from Reviewers:

Reviewer #2: The authors have addressed all my points.

I noticed a typo, the R package is called metafor not metaphor.

As far as the editorial request for the test underlying the p-values in the results since the authors used REML they cannot be from LRTs so they must be Wald type tests for the moderator variable. That is certainly what metafor produces (I do not use meta).

Michael Dewey

[LINK]

---

## [Editor Report · Decision Letter 2]

7 Nov 2019

Dear Dr Woodd, 

On behalf of my colleagues and the academic editor, Dr. Gordon C. Smith, I am delighted to inform you that your manuscript entitled "Incidence of maternal peripartum infection: A systematic review and meta-analysis" (PMEDICINE-D-19-02904R2) has been accepted for publication in PLOS Medicine. 

PRODUCTION PROCESS

PRESS

PROFILE INFORMATION

Thank you again for submitting the manuscript to PLOS Medicine. We look forward to publishing it. 

Best wishes, 

Clare Stone, PhD

Managing Editor 

PLOS Medicine

plosmedicine.org